# One-step Latent-free Image Generation with Pixel Mean Flows

Yiyang Lu [* 1]   Susie Lu [* 1]   Qiao Sun [* 1]   Hanhong Zhao [* 1]   Zhicheng Jiang [1]   Xianbang Wang [1]
Tianhong Li [1]   Zhengyang Geng [2]   Kaiming He [1]

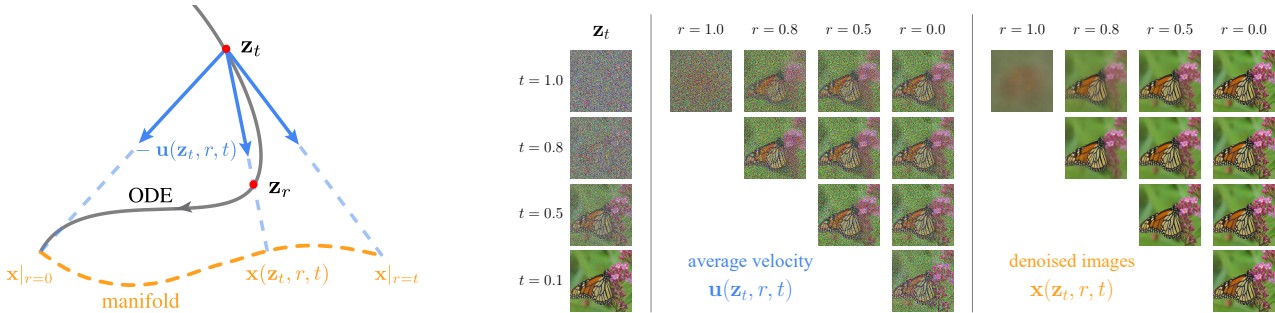

*Figure 1.* **The pixel MeanFlow (pMF) formulation, driven by the manifold hypothesis. (Left)**: Following MeanFlow (Geng et al., 2025a), pMF aims to approximate the *average velocity field* $\mathbf{u}(\mathbf{z}_t, r, t)$ induced by the underlying ODE trajectory (black). We define a new field $\mathbf{x}(\mathbf{z}_t, r, t) \triangleq \mathbf{z}_t - t \cdot \mathbf{u}(\mathbf{z}_t, r, t)$, which behaves like denoised images. We hypothesize that $\mathbf{x}$ approximately lies on a low-dimensional data manifold (orange curve) and can therefore be more accurately approximated by a neural network. **(Right)**: Visualization of the quantities $\mathbf{z}_t$, $\mathbf{u}$, $\mathbf{x}$ obtained by tracking an ODE trajectory via simulation. The average velocity field $\mathbf{u}$ corresponds to *noisy* images and is inevitably higher-dimensional; the induced field $\mathbf{x}$ corresponds to approximately clean or blurred images, which can be easier to model by a neural network.

## Abstract

Modern diffusion/flow-based models for image generation typically exhibit two core characteristics: (i) using *multi-step* sampling, and (ii) operating in a *latent* space. Recent advances have made encouraging progress on each aspect individually, paving the way toward *one-step diffusion/flow without latents*. In this work, we take a further step towards this goal and propose "*pixel MeanFlow*" (pMF). Our core guideline is to formulate the network output space and the loss space separately. The network target is designed to be on a presumed low-dimensional image manifold (*i.e.*, $\mathbf{x}$-prediction), while the loss is defined via MeanFlow in the velocity space. We introduce a simple transformation between the image manifold and the average velocity field. In experiments, pMF achieves strong results for one-step latent-free generation on ImageNet at 256×256 resolution (2.22 FID) and 512×512 resolution (2.48 FID), filling a key missing piece in this regime. We hope that our study will further

advance the boundaries of diffusion/flow-based generative models.

## 1. Introduction

Modern diffusion/flow-based models for image generation are largely characterized by two core aspects: (i) using *multi-step* sampling (Sohl-Dickstein et al., 2015), and (ii) operating in a *latent* space (Rombach et al., 2022). Both aspects concern decomposing a highly complex generation problem into more tractable subproblems. While these have been the commonly used solutions, it is valuable, from both scientific and efficiency perspectives, to investigate alternatives that do not rely on these components.

The community has made encouraging progress on each of the two aspects *individually*. On one hand, Consistency Models (Song et al., 2023) and subsequent developments, *e.g.*, MeanFlow (MF) (Geng et al., 2025a), have substantially advanced few-/one-step sampling. On the other hand, there have been promising advances in image generation in the raw pixel space, *e.g.*, using "Just image Transformers" (JiT) (Li & He, 2025). Taken together, it appears that the community is now equipped with the key ingredients for one-step latent-free generation.

However, merging these two separate directions poses a more demanding task for the neural network, which should not be assumed to have infinite capacity in practice. On one

*Equal contribution  [1]MIT  [2]CMU. Correspondence to: Kaiming He <kaiming@mit.edu>.

*Proceedings of the 43rd International Conference on Machine Learning*, Seoul, South Korea. PMLR 306, 2026. Copyright 2026 by the author(s).

hand, in few-step modeling, a single network is responsible for modeling trajectories across different start and end points; on the other hand, in the pixel space, the model must explicitly or implicitly perform compression and abstraction (*i.e.*, manifold learning) in the absence of pre-trained latent tokenizers. Given the challenges posed by each individual issue, it is nontrivial to design a unified network that simultaneously satisfies properties of both aspects.

In this work, we propose *pixel MeanFlow* (pMF) for one-step latent-free image generation. pMF follows the improved MeanFlow (iMF) (Geng et al., 2025b) that learns the average velocity field (namely, $\mathbf{u}$) using a loss defined in the space of instantaneous velocity (namely, $\mathbf{v}$). On the other hand, following JiT (Li & He, 2025), pMF directly parameterizes a denoised-image-like quantity (namely, $\mathbf{x}$-prediction), which is expected to lie on a low-dimensional manifold. To accommodate both formulations, we introduce a conversion that relates the fields $\mathbf{v}$, $\mathbf{u}$, and $\mathbf{x}$. We empirically show that this formulation better aligns with the manifold hypothesis (Chapelle et al., 2006) and yields a more learnable target (see Fig .1).

Generally speaking, pMF learns a network that directly maps noisy inputs to image pixels. It enables a "what-you-see-is-what-you-get" property, which is not the case for multi-step or latent-based methods. This property makes the usage of the perceptual loss (Zhang et al., 2018) a natural component for pMF, further enhancing generation quality.

Experimental results show that pMF performs strongly for one-step latent-free generation, reaching 2.22 FID at 256×256 and 2.48 FID at 512×512 on ImageNet (Deng et al., 2009). We further demonstrate that a proper prediction target (Chapelle et al., 2006) is critical: directly predicting a velocity field in pixel space leads to catastrophic performance. Our study reveals that one-step latent-free generation is becoming both feasible and competitive, marking a solid step toward *direct generative modeling* formulated as a single, end-to-end neural network.

## 2. Related Work

**Diffusion and Flow Matching.** Diffusion models (Sohl-Dickstein et al., 2015; Ho et al., 2020; Song et al., 2021a) and Flow Matching (Lipman et al., 2023; Liu et al., 2023; Albergo & Vanden-Eijnden, 2023) have become cornerstones of modern generative modeling. These approaches can be formulated as learning probability flows that transform one distribution into another. During inference, samples are generated by solving differential equations (SDEs/ODEs), often through a numerical solver with multiple function evaluations.

In today's practice, diffusion/flow-based methods often operate in a latent space (Rombach et al., 2022). The latent

tokenizer substantially reduces the dimensionality of the space, while enabling a focus on high-level semantics (via the perceptual loss (Zhang et al., 2018)) and forgiving low-level nuance (via the adversarial loss (Goodfellow et al., 2014)). Latent-based methods have become the standard choice for high-resolution image generation (Rombach et al., 2022; Peebles & Xie, 2023; Ma et al., 2024).

**Pixel-space Diffusion and Flows.** Before the prevalence of using latents, diffusion models were originally developed in the pixel-space (Ho et al., 2020; Song et al., 2021b; Nichol & Dhariwal, 2021; Dhariwal & Nichol, 2021). These methods are in general based on a U-net structure (Ronneberger et al., 2015), which, unlike Vision Transformers (ViT) (Dosovitskiy et al., 2021), does not rely on aggressive patchification.

There has been a recent trend in investigating pixel-space *Transformer* models for diffusion and flows (Chen et al., 2025a; Wang et al., 2025; Lei et al., 2026; Li & He, 2025; Yu et al., 2025b; Ma et al., 2025; Chen et al., 2025b). To address the high dimensionality of the *patch* space, a series of work focuses on designing a "refiner head" that covers the details lost in patch-based Transformers. Another solution, proposed in JiT (Li & He, 2025), predicts the denoised image (*i.e.*, $\mathbf{x}$) that is hypothesized to lie on a low-dimensional manifold (Chapelle et al., 2006).

**One-step Diffusion and Flows.** It is of both practical and theoretical interest to study reducing steps in diffusion/flow-based models. Early explorations (Salimans & Ho, 2022; Meng et al., 2023) along this direction rely on distilling pretrained multi-step models into few-step variants. Consistency Models (CM) (Song et al., 2023) demonstrate that it is possible to train one-step models from scratch. CM and its improvements (Song et al., 2024; Geng et al., 2024; Lu & Song, 2025) aim to learn a network that maps any point along the ODE trajectory to its end point.

A series of one-step models (Kim et al., 2024; Boffi et al., 2025; Frans et al., 2025; Zhou et al., 2025; Geng et al., 2025a;b) have been developed to characterize SDE/ODE trajectories. Conceptually, these methods predict a quantity that depends on two time steps along a trajectory. The designs of these different methods typically differ in *what quantity is to be predicted*, as well as in how the quantity of interest is characterized by a loss function. Our method addresses these issues too. We provide detailed discussions in context later (Sec. 4.5).

## 3. Background

Our pMF is built on top of Flow Matching (Lipman et al., 2023; Liu et al., 2023; Albergo & Vanden-Eijnden, 2023), MeanFlow (Geng et al., 2025a;b), and JiT (Li & He, 2025). We briefly introduce the background as follows.

*Table 1.* **Prediction space and loss space.** Here, all methods are Transformer-based. The notations include noise $\epsilon$, data $\mathbf{x}$, instantaneous velocity $\mathbf{v}$, and average velocity $\mathbf{u}$. Prediction space is that of the direct output of the network; loss space is that of the regression target. When the prediction and loss spaces do not match, a space conversion is introduced. Here, the compared methods are: DiT (Peebles & Xie, 2023), SiT (Ma et al., 2024), MeanFlow (MF) (Geng et al., 2025a), improved MF (iMF) (Geng et al., 2025b), and JiT (Li & He, 2025).

|      | pred. space | conversion | loss space |
|------|:-----------:|:----------:|:----------:|
| DiT  | $\epsilon$  | -          | $\epsilon$ |
| SiT  | $\mathbf{v}$ | -         | $\mathbf{v}$ |
| MF   | $\mathbf{u}$ | -         | $\mathbf{u}$ |
| iMF  | $\mathbf{u}$ | $\mathbf{u} \to \mathbf{v}$ | $\mathbf{v}$ |
| JiT  | $\mathbf{x}$ | $\mathbf{x} \to \mathbf{v}$ | $\mathbf{v}$ |
| **pMF** | $\mathbf{x}$ | $\mathbf{x} \to \mathbf{u} \to \mathbf{v}$ | $\mathbf{v}$ |

**Flow Matching.** Flow Matching (FM) learns a velocity field $\mathbf{v}$ that maps a prior distribution $p_{\text{prior}}$ to the data distribution $p_{\text{data}}$. We consider the standard linear interpolation schedule:

$$\mathbf{z}_t = (1 - t)\mathbf{x} + t\epsilon \tag{1}$$

with data $\mathbf{x} \sim p_{\text{data}}$ and noise $\epsilon \sim p_{\text{prior}}$ (*e.g.*, Gaussian), and time $t \in [0, 1]$. At $t = 0$, there is: $\mathbf{z}_0 \sim p_{\text{data}}$. The interpolation yields a *conditional* velocity $\mathbf{v} \triangleq \mathbf{z}_t'$:

$$\mathbf{v} = \epsilon - \mathbf{x} \tag{2}$$

FM optimizes a network $\mathbf{v}_\theta$, parameterized by $\theta$, by minimizing a loss function in the $\mathbf{v}$-space (namely, "$\mathbf{v}$-loss"):

$$\mathcal{L}_{\text{FM}} = \mathbb{E}_{t,\mathbf{x},\epsilon}\|\mathbf{v}_\theta(\mathbf{z}_t, t) - \mathbf{v}\|^2. \tag{3}$$

It is shown (Lipman et al., 2023) that the underlying target of $\mathbf{v}_\theta$ is the *marginal* velocity $\mathbf{v}(\mathbf{z}_t, t) \triangleq \mathbb{E}[\mathbf{v}|\mathbf{z}_t, t]$.

At inference time, samples are generated by solving the ODE: $\mathrm{d}\mathbf{z}_t/\mathrm{d}t = \mathbf{v}_\theta(\mathbf{z}_t, t)$, from $t = 1$ to $t = 0$, with $\mathbf{z}_1 = \epsilon \sim p_{\text{prior}}$. This can be done by numerical methods such as Euler or Heun-based solvers.

**Flow Matching with x-prediction.** The quantity $\mathbf{v}$ in Eq. (2) is a noisy image. To facilitate the usage of Transformers operated on pixels, JiT (Li & He, 2025) opts to parameterize the data $\mathbf{x}$ by the neural network and convert it to velocity $\mathbf{v}$ by:[1]

$$\mathbf{v}_\theta(\mathbf{z}_t, t) := \frac{1}{t}(\mathbf{z}_t - \mathbf{x}_\theta(\mathbf{z}_t, t)), \tag{4}$$

where $\mathbf{x}_\theta = \texttt{net}_\theta$ is the direct output of a Vision Transformer (ViT) (Dosovitskiy et al., 2021). This formulation is referred to as **x-prediction**, whereas the **v-loss** in Eq. (2) is used for training. Tab. 1 lists the relation.

_______________

[1] In JiT (Li & He, 2025), $t = 0$ corresponds to the noise side, in contrast to our convention of $t = 1$. Their convention leads to a coefficient of $\frac{1}{1-t}$, rather than $\frac{1}{t}$ here.

**Mean Flows.** The MeanFlow (MF) framework (Geng et al., 2025a) learns an *average velocity* field $\mathbf{u}$ for few-/one-step generation. With FM's $\mathbf{v}$ viewed as the *instantaneous velocity*, MF defines the average velocity $\mathbf{u}$ as:

$$\mathbf{u}(\mathbf{z}_t, r, t) \triangleq \frac{1}{t - r} \int_r^t \mathbf{v}(\mathbf{z}_\tau, \tau)\mathrm{d}\tau, \tag{5}$$

where $r$ and $t$ are two time steps: $0 \leq r \leq t \leq 1$. This definition leads to a *MeanFlow Identity* (Geng et al., 2025a;b):

$$\mathbf{v}(\mathbf{z}_t, t) = \mathbf{u}(\mathbf{z}_t, r, t) + (t - r)\frac{\mathrm{d}}{\mathrm{d}t}\mathbf{u}(\mathbf{z}_t, r, t), \tag{6}$$

This identity provides a way for defining a prediction function with a network $\mathbf{u}_\theta$ (Geng et al., 2025b):

$$\mathbf{V}_\theta \triangleq \mathbf{u}_\theta + (t - r) \cdot \text{JVP}_{\text{sg}}. \tag{7}$$

Here, the capital $\mathbf{V}_\theta$ corresponds to the left-hand side of Eq. (6), and on the right-hand side, JVP denotes the Jacobian-vector product for computing $\frac{\mathrm{d}}{\mathrm{d}t}\mathbf{u}_\theta$, with "sg" denoting stop-gradient. We follow the JVP computation and implementation of iMF (Geng et al., 2025b), which is not the focus of our paper. With the definition in Eq. (7), iMF minimizes the $\mathbf{v}$-loss like Eq. (3), *i.e.*, $\|\mathbf{V}_\theta - \mathbf{v}\|^2$. This formulation can be viewed as **u-prediction** with **v-loss** (see also Tab. 1).

## 4. Pixel MeanFlow

To facilitate one-step, latent-free generation, we introduce pixel MeanFlow (pMF). The core design of pMF is to establish a connection between the different fields of $\mathbf{u}$, $\mathbf{v}$, and $\mathbf{x}$. We want the network to directly output $\mathbf{x}$, like JiT (Li & He, 2025), whereas one-step modeling is performed on the space of $\mathbf{u}$ and $\mathbf{v}$ as in MeanFlow (Geng et al., 2025a;b).

### 4.1. The Denoised Image Field

As discussed in Sec. 3, both iMF (Geng et al., 2025b) and JiT (Li & He, 2025) can be viewed as minimizing the **v-loss**, while iMF performs **u-prediction** and JiT performs **x-prediction**. Accordingly, we introduce a connection between $\mathbf{u}$ and a generalized form of $\mathbf{x}$.

Consider the average velocity field $\mathbf{u}$ defined in Eq. (5): this field represents an underlying *ground-truth* quantity that depends on $p_{\text{data}}$, $p_{\text{prior}}$, and the time schedule, but *not* on the network (and thus has no dependence on parameters $\theta$). We induce a new field $\mathbf{x}(\mathbf{z}_t, r, t)$ defined as:

$$\boxed{\mathbf{x}(\mathbf{z}_t, r, t) \triangleq \mathbf{z}_t - t \cdot \mathbf{u}(\mathbf{z}_t, r, t).} \tag{8}$$

As detailed below, this field $\mathbf{x}$ serves a role similar to *denoised images*. Unlike other quantities that are sometimes

referred to as "$\mathbf{x}$" in prior works, our field $\mathbf{x}(\mathbf{z}_t, r, t)$ is indexed by two time steps, $r$ and $t$: for any given $\mathbf{z}_t$, our $\mathbf{x}$ is a 2D field indexed by $(r, t)$, rather than a 1D trajectory indexed only by $t$.

### 4.2. The Generalized Manifold Hypothesis

Fig. 1 visualizes the field of $\mathbf{u}$ and the field of $\mathbf{x}$ by simulating one ODE trajectory obtained from a pretrained FM model. As illustrated, $\mathbf{u}$ consists of *noisy* images, because, as a velocity field, $\mathbf{u}$ contains both noise and data components. In contrast, the field $\mathbf{x}$ has the appearance of denoised images: they are nearly clean images, or overly denoised images that appear blurry. Next, we discuss how the manifold hypothesis can be generalized to this quantity $\mathbf{x}$.

Note that the time step $r$ in MF satisfies: $0 \le r \le t$. We first show that the boundary cases at $r = t$ and $r = 0$ can approximately satisfy the manifold hypothesis; we then discuss the case $0 < r < t$.

**Boundary case I: $r = t$.** When $r = t$, the average velocity $\mathbf{u}$ degenerates to the instantaneous velocity $\mathbf{v}$, *i.e.*, $\mathbf{u}(\mathbf{z}_t, t, t) = \mathbf{v}(\mathbf{z}_t, t)$. In this case, Eq. (8) gives us:

$$\mathbf{x}(\mathbf{z}_t, t, t) = \mathbf{z}_t - t \cdot \mathbf{v}(\mathbf{z}_t, t). \qquad (9)$$

This is essentially the $\mathbf{x}$-prediction target used in JiT (Li & He, 2025): see Eq. (4). Intuitively, this $\mathbf{x}$ is the denoised image to be predicted by JiT. This denoised image can be blurry if the noise level is high (as it should be the expectation of different image samples that can produce the same noisy data $\mathbf{z}_t$). As widely observed in classical image denoising research, these denoised images can be assumed as approximately on a low-dimensional (or *lower*-dimensional) manifold (Vincent et al., 2008). See the images corresponding to $r = t$ in Fig. 1(right).

**Boundary case II: $r = 0$.** The definition of $\mathbf{u}$ in Eq. (5) gives: $\mathbf{u}(\mathbf{z}_t, 0, t) = \frac{1}{t} \int_0^t \mathbf{v}(z_\tau, \tau) d\tau = \frac{1}{t}(\mathbf{z}_t - \mathbf{z}_0)$. Substituting it into Eq. (8) gives:

$$\mathbf{x}(\mathbf{z}_t, 0, t) = \mathbf{z}_0, \qquad (10)$$

*i.e.*, it is the endpoint of the ODE trajectory. For a ground-truth ODE trajectory, there is: $\mathbf{z}_0 \sim p_{\text{data}}$, that is, it should follow the image distribution. Therefore, we can assume that $\mathbf{x}(\mathbf{z}_t, 0, t)$ is approximately on the image manifold.

**General case: $r \in (0, t)$.** Unlike the boundary cases, the quantity $\mathbf{x}(\mathbf{z}_t, r, t)$ is not guaranteed to correspond to an (possibly blurry) image sample from the data manifold. Nevertheless, empirically, our simulations (Fig. 1, right) suggest that $\mathbf{x}$ appears like a denoised image. It stands in sharp contrast to velocity-space quantities ($\mathbf{u}$ in Fig. 1), which are significantly noisier. This comparison suggests that $\mathbf{x}$ may be easier to model by a neural network than the noisier

---

**Algorithm 1** pixel MeanFlow: training.

Note: in PyTorch and JAX, `jvp` returns the function output and JVP.

```
# net: x-prediction network
# x: training batch in pixels

t, r = sample_t_r()
e = randn_like(x)

z = (1 - t) * x + t * e

# average velocity u from x-prediction
def u_fn(z, r, t):
    return (z - net(z, r, t)) / t

# instantaneous velocity v at time t
v = u_fn(z, t, t)

# predict u and dudt
u, dudt = jvp(u_fn, (z, r, t), (v, 0, 1))

# compound function V
V = u + (t - r) * stopgrad(dudt)

loss = metric(V, e - x)
```

$\mathbf{u}$. Our experiments in Sec. 5 and Sec. 6 show that, for our pixel-space model, $\mathbf{x}$-prediction performs effectively, whereas $\mathbf{u}$-prediction degrades severely.

### 4.3. Algorithm

The induced field $\mathbf{x}$ in Eq. (8) provides a re-parameterization of the MeanFlow network. Specifically, we let the network $\text{net}_\theta$ directly output $\mathbf{x}$, and compute the corresponding velocity field $\mathbf{u}$ via Eq. (8) as

$$\mathbf{u}_\theta(\mathbf{z}_t, r, t) = \frac{1}{t}\big(\mathbf{z}_t - \mathbf{x}_\theta(\mathbf{z}_t, r, t)\big). \qquad (11)$$

Here, $\mathbf{x}_\theta(\mathbf{z}_t, r, t) := \text{net}_\theta(\mathbf{z}_t, r, t)$ is the direct output of the network, following JiT. This formulation is a natural extension of Eq. (4).

We incorporate $\mathbf{u}_\theta$ in (11) into the iMF formulation (Geng et al., 2025b), *i.e.*, using Eq. (7) with $\mathbf{v}$-loss. Specifically, our optimization objective is:

$$\mathcal{L}_{\text{pMF}} = \mathbb{E}_{t, r, \mathbf{x}, \boldsymbol{\epsilon}} \|\mathbf{V}_\theta - \mathbf{v}\|^2, \qquad (12)$$
$$\text{where} \quad \mathbf{V}_\theta \triangleq \mathbf{u}_\theta + (t - r) \cdot \text{JVP}_{\text{sg}}.$$

Conceptually, this is $\mathbf{v}$-loss with $\mathbf{x}$-prediction, while $\mathbf{x}$ is converted to the $\mathbf{v}$-space by the relation of $\mathbf{x} \to \mathbf{u} \to \mathbf{V}$ for regressing $\mathbf{v}$. Tab. 1 summarizes the relation.

The corresponding pseudo-code is in Alg. 1. Following iMF (Geng et al., 2025b), this algorithm can be extended to support CFG (Ho & Salimans, 2021), which we omit here for brevity and we elaborate on in the appendix.

## 4.4. Pixel MeanFlow with Perceptual Loss

The network $\mathbf{x}_\theta(\mathbf{z}_t, r, t)$ directly maps a noisy input $\mathbf{z}_t$ to a denoised image. This enables a "what-you-see-is-what-you-get" behavior at training time. Accordingly, in addition to the $\ell_2$ loss, we can further incorporate the perceptual loss (Zhang et al., 2018). Latent-based methods (Rombach et al., 2022) benefit from perceptual losses during tokenizer reconstruction training, whereas pixel-based methods have not readily leveraged this benefit.

Formally, as $\mathbf{x}_\theta$ is a denoised image in pixels, we directly apply the perceptual loss (*e.g.*, LPIPS (Zhang et al., 2018)) on it. Our overall training objective is $\mathcal{L} = \mathcal{L}_{\text{pMF}} + \lambda \mathcal{L}_{\text{perc}}$, where $\mathcal{L}_{\text{perc}}$ denotes the perceptual loss between $\mathbf{x}_\theta$ and the ground-truth clean image $\mathbf{x}$, and $\lambda$ is a weight hyper-parameter. In practice, the perceptual loss can be applied only when the added noise is below a certain threshold (*i.e.*, $t \le t_{\text{thr}}$), such that the denoised image is not too blurry.

We investigate the standard LPIPS loss based on the VGG classifier (Simonyan & Zisserman, 2015) and a variant based on ConvNeXt-V2 (Woo et al., 2023) (see Appendix A).

## 4.5. Relation to Prior Works

Our pMF is closely related to several prior few-/one-step methods, which we discuss next. The relations and differences involve the prediction target and training formulation.

**Consistency Models (CM)** (Song et al., 2023; 2024; Geng et al., 2024; Lu & Song, 2025) learn a mapping from a noisy sample $\mathbf{z}_t$ directly to a generated image. In our notation, this corresponds to *fixing* the endpoint to $r = 0$. In our $(r, t)$-coordinate plane, this amounts to sampling along the line of $r = 0$ for any $t$.

In addition, while consistency models aim to predict an image, they often employ a pre-conditioner (Karras et al., 2022) that modifies the underlying prediction target. In our notation, their $\mathbf{x}_\theta$ has a form of $\mathbf{x}_\theta := c_{\text{skip}} \cdot \mathbf{z}_t + c_{\text{out}} \cdot \texttt{net}_\theta$. Unless $c_{\text{skip}}$ is zero, the network does *not* perform $\mathbf{x}$-prediction. We provide ablation study in experiments.

**Consistency Trajectory Models** (CTM) (Kim et al., 2024) formulate a two-time quantity and enable flexible $(r, t)$-plane modeling. Unlike MeanFlow, which is based on a derivative formulation, CTM relies on integrating the ODE during training. Besides, CTM adopts a pre-conditioner, similar to CM, and therefore does not directly output the image through the network.

**Flow Map Matching** (FMM) (Boffi et al., 2025) is also based on a two-time quantity (referred to as a Flow Map), for which several training objectives have been developed. In our notation, the Flow Map plays a role like *displacement*, *i.e.*, $\mathbf{z}_t - \mathbf{z}_r$. This quantity generally does not lie on a low-dimensional manifold (*e.g.*, $\mathbf{z}_1 - \mathbf{z}_0$ is a noisy image), and a

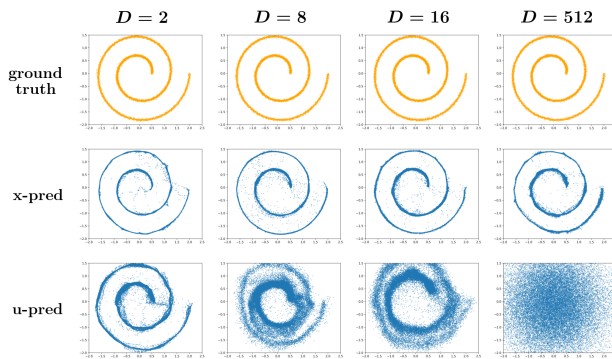

*Figure 2.* **Toy Experiment.** A 2D toy dataset is linearly projected into a $D$-dimensional observation space using a fixed, $D \times 2$ column-orthonormal projection matrix. We train Mean-Flow models with either the original $\mathbf{u}$-prediction or the proposed $\mathbf{x}$-prediction, for $D \in \{2, 8, 16, 512\}$. We visualize 1-NFE generation results. The models use the same 7-layer ReLU MLP backbone with 256 hidden units. The $\mathbf{x}$-prediction formulation produces reasonably good results, whereas $\mathbf{u}$-prediction fails in the case of high-dimensional observation spaces.

further re-parameterization may be desired in the demanding scenario considered in this paper.

## 5. Toy Experiments

We demonstrate with a 2D toy experiment (Fig. 2) that $\mathbf{x}$-prediction is preferable in MeanFlow when the underlying data lie on a low-dimensional manifold. The experimental setting follows the one in Li & He (2025).

Formally, we consider an underlying data distribution (here, Swiss roll) defined on a 2D space. The data is projected into a $D$-dimensional *observation* space using a $D \times 2$ column-orthogonal matrix. We train MeanFlow models on the $D$-dim observation space, for $D \in \{2, 8, 16, 512\}$. We compare the $\mathbf{u}$-prediction in Geng et al. (2025b) with our $\mathbf{x}$-prediction. The network is a 7-layer ReLU MLP with 256 hidden units.

Fig. 2 shows that $\mathbf{x}$-prediction performs reasonably well, whereas $\mathbf{u}$-prediction degrades rapidly when $D$ increases. We observe that this performance gap is reflected by the differences in the training loss (noting that both minimize the same $\mathbf{v}$-loss): $\mathbf{x}$-prediction yields lower training loss than the $\mathbf{u}$-prediction counterpart. This suggests that predicting $\mathbf{x}$ is easier for a network with limited capacity.

## 6. ImageNet Experiments

We conduct ablation on ImageNet (Deng et al., 2009) at resolution $256 \times 256$ by default. We report Fréchet Inception Distance (FID; Heusel et al. (2017)) on 50,000 generated samples. All of our models generate *raw pixel* images with a *single* function evaluation (1-NFE).

*Table 2.* **x-prediction is crucial for high-dimensional pixel-space generation.** We compare **x**- and **u**-prediction on ImageNet using a fixed sequence length of $16^2$. **(a)**: At **64×64** resolution, the patch dimension is 48 (4×4×3). Both prediction targets work well. **(b)**: At **256×256** resolution, the patch dimension is 768 (16×16×3). **u**-prediction fails catastrophically, whereas **x**-prediction performs reasonably well. This baseline (with 9.56 FID) is our ablation setting. For fair comparison, no bottleneck embedding (Li & He, 2025) is adopted in our ablation. (Settings: Muon optimizer, MSE loss, 160 epochs).

|  | img size | model arch | patch size | seq len | **patch dim** | **1-NFE FID** | |
|---|---|---|---|---|---|---|---|
|  |  |  |  |  |  | **x-pred** | **u-pred** |
| (a) | 64×64 | B/4 | $4^2$ | $16^2$ | 48 | 3.80 | 3.82 |
| (b) | 256×256 | B/16 | $16^2$ | $16^2$ | 768 | 9.56 | 164.89 |

We adopt the iMF architecture (Geng et al., 2025b), which is a variant of the DiT design (Peebles & Xie, 2023). Unless specified, we set the patch size to 16×16 (denoted as pMF/16). Ablation models are trained *from scratch* for 160 epochs. More details are in Appendix A.

### 6.1. Prediction Targets of the Network

Our method is based on the manifold hypothesis, which assumes that **x** is in a low-dimensional manifold and easier to predict. We verify this assumption in Tab. 2.

First, we consider the case of 64×64 resolution as a simpler setting. With a patch size of 4×4, the patch dimension is 48 (4×4×3). This dimensionality is substantially lower than the network capacity (hidden dimension 768). As a result, pMF performs well under both **x**- and **u**-prediction.

Next, we consider the case of 256×256 resolution. With a patch size of 16×16, as common practice, the patch dimension is 768 (16×16×3). This leads to a high-dimensional observation space that is more difficult for a neural network to model. In this case, only **x**-prediction performs well, suggesting that **x** is on a lower-dimensional manifold and is therefore more amenable to learning. In contrast, **u**-prediction fails catastrophically: as a noisy quantity, **u** has full support in the high-dimensional space and is much harder to model. These observations are consistent with those in Li & He (2025).

### 6.2. Ablations Studies

We further ablate other important factors, discussed next.

**Optimizer.** We find that the choice of optimizer plays an important role in pMF. In Fig. 3a, we compare the standard Adam optimizer (Kingma & Ba, 2015) with the recently proposed **Muon** (Jordan et al., 2024). Muon exhibits faster convergence and substantially improved FID.

In our preliminary experiments, we compared Adam with Muon on *multi-step* diffusion: while Muon exhibits faster convergence, we did not observe a final improvement. This

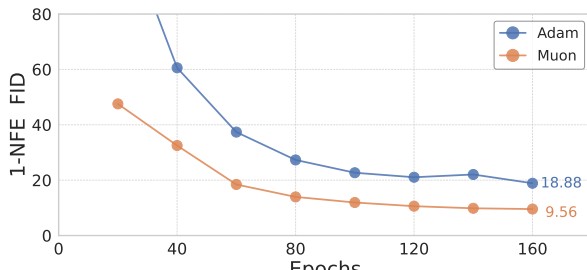

*(a)* **Muon *vs.* Adam.** Muon converges faster and achieves better FID. At 320 epochs, Adam reaches 11.86 FID, while Muon achieves 8.71 FID. (Settings: pMF-B/16, MSE loss)

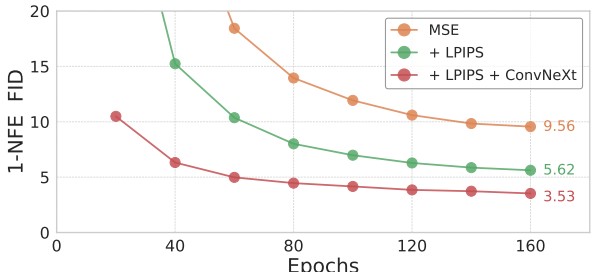

*(b)* **Perceptual loss.** Using standard VGG-based LPIPS as well as a ConvNeXt-based variant leads to improved FID. (Settings: pMF-B/16, Muon optimizer)

*Figure 3.* **Training curves of pMF** on ImageNet 256×256 with pixel-space, 1-NFE generation.

suggests that the benefit of faster convergence is more pronounced in our *single-step* setting. In MeanFlow, the stop-gradient target (*e.g.*, Eq. (12)) depends on the network evaluation, and a better network in early epochs (enabled by Muon) can provide a more accurate target. Accordingly, the benefit of faster convergence is further amplified.

**Perceptual loss.** Thus far, our ablation studies are conducted using a simple $\ell_2$ loss. In Fig. 3b, we further incorporate perceptual loss. Using the standard VGG-based LPIPS (Zhang et al., 2018) improves FID from 9.56 to 5.62; incorporating a ConvNeXt-V2 variant (Woo et al., 2023) further improves FID to 3.53. Overall, incorporating perceptual loss leads to an improvement of about 6 FID points.

In standard latent-based methods (Rombach et al., 2022), perceptual loss plays a key role in training the VAE tokenizer (often in conjunction with an adversarial loss, which we do not investigate). We note that the VAE decoder directly outputs a reconstructed image (*i.e.*, **x**) in pixel space, making the use of perceptual loss amenable. As our generator likewise outputs **x** in pixel space in one step, it naturally benefits from the same property.

**Alternative: pre-conditioner.** Pre-conditioners (Karras et al., 2022) have been a common strategy for re-parameterizing the predict target. Using our notation, a pre-conditioner performs: $\mathbf{x}_\theta = c_{\text{skip}} \cdot \mathbf{z}_t + c_{\text{out}} \cdot \texttt{net}_\theta$.

*Table 3.* **Alternative designs of pMF**, evaluated on ImageNet 256×256 with pixel-space, 1-NFE generation. (Settings: pMF-B/16, Muon optimizer, w/ perceptual loss, 160 epochs)

| | pre-conditioned **x**-pred | | | **x**-pred |
|---|---|---|---|---|
| | linear | EDM-style | sCM-style | (no pre-cond) |
| **1-NFE FID** | 34.61 | 14.43 | 13.81 | **3.53** |

*(a)* **Comparison with pre-conditioners.** A pre-conditioner transforms the direct network output into **x**, and may therefore cause it to deviate from a low-dimensional manifold.

| time sampler | **1-NFE FID** |
|---|---|
| only $r = t$ | 194.53 |
| only $r = 0$ | 389.28 |
| only $r = t$ and $r = 0$ | 106.59 |
| $0 \leq r \leq t$ (ours) | **3.53** |

*(b)* **Comparison on time samplers**. Our method, following Mean-Flow, performs time sampling in the $(r, t)$-coordinate plane. Our sampler covers the full region in $0 \leq r \leq t$. Restricting to a single line ($r = t$, or $r = 0$) or to both lines leads to failure.

We compare three variants of pre-conditioners: (i) linear ($c_{\text{skip}} = 1 - t$, $c_{\text{out}} = t$); (ii) the EDM style (Karras et al., 2022); and (iii) the sCM style (Lu & Song, 2025).

Tab. 3a compares the pre-conditioners used in place of pMF's **x**-prediction. Both the EDM- and sCM-style pre-conditioners outperform a naive linear variant, suggesting that performance depends strongly on the choice of parameterization. However, in the very high-dimensional input regime considered here, our simple **x**-prediction is preferable and achieves better performance. This is because, unless $c_{\text{skip}} = 0$, the network prediction deviates from the **x**-space and may lie on a higher-dimensional manifold.

**Alternative: time samplers.** Our method performs time sampling in the $(r, t)$-coordinate plane. We study alternative designs that restrict time sampling to specific cases: **(i)** only $r = t$, which amounts to Flow Matching; **(ii)** only $r = 0$, which conceptually analogize the CM (Song et al., 2023) regime; and **(iii)** a combination of both.

Tab. 3b shows the results of these restricted time samplers. None of these alternatives is sufficient to address the challenging scenario considered here. This comparison suggests that MeanFlow methods leverage the relations across $(r, t)$ points to learn the field, and restricting time sampling to one or two lines may undermine this formulation.

**High-resolution generation.** In Tab. 4, we investigate pMF at resolution 256, 512, and 1024. We keep the sequence length unchanged ($16^2$), thereby roughly maintaining the computational cost across different resolutions. Doing so leads to an aggressively large patch size (*e.g.*, $64^2$) and patch dimensionality (*e.g.*, 12288).

Tab. 4 shows that pMF can effectively handle this highly challenging case. Even though the observation space is high-dimensional, our model always predicts **x**, whose un-

*Table 4.* **High-resolution generation** on ImageNet. We fix sequence length ($16^2$) by increasing patch size, pMF performs strongly despite the extremely high per-patch dimensionality. (Settings: Muon optimizer, w/ perceptual loss, 160 epochs)

| img size | model arch | patch size | seq len | **patch dim** | hidden dim | **1-NFE FID** |
|---|---|---|---|---|---|---|
| 256×256 | B/16 | $16^2$ | $16^2$ | 768 | 768 | 3.53 |
| 512×512 | B/32 | $32^2$ | $16^2$ | 3072 | 768 | 4.06 |
| 1024×1024 | B/64 | $64^2$ | $16^2$ | 12288 | 768 | 4.58 |

*Table 5.* **Scalability.** Increasing the model size and training epochs improves results. (Settings: Muon optimizer, w/ perceptual loss)

| | depth | width | # params | Gflops | **1-NFE FID** | |
|---|---|---|---|---|---|---|
| | | | | | 160-ep | 320-ep |
| B/16 | 16 | 768 | 119 | 34 | 3.53 | 3.12 |
| L/16 | 32 | 1024 | 411 | 117 | 2.85 | 2.52 |
| H/16 | 48 | 1280 | 956 | 271 | 2.57 | 2.29 |

derlying dimensionality does not grow proportionally. This enables a highly FLOP-efficient solution for high-resolution generation, *e.g.*, as will be shown in Tab. 7 at 512×512.

**Scalability.** In Tab. 5, we report results of increasing the model size and training epochs. As expected, pMF benefits from scaling along both axes. Qualitative examples are provided in Fig. 4 and Appendix B.

### 6.3. System-level Comparisons

We compare with previous methods in Tab. 6 (256×256) and Tab. 7 (512×512). Given that few existing methods are *both* one-step and latent-free, we include multi-step and/or latent-based methods for reference. We consider methods that are trained *from scratch*, without distillation.

**ImageNet 256×256.** Tab. 6 shows that our method achieves 2.22 (at 360 epochs). To our knowledge, the only other method in this category (one-step, latent-free diffusion/flow) is the recently proposed EPG (Lei et al., 2026), which reaches 8.82 FID with self-supervised pre-training.

GANs (Goodfellow et al., 2014) are another category of methods that are competitive for one-step, latent-free generation. In comparison with the leading GAN results, our pMF achieves comparable FID with substantially lower compute, as well as better scalability. In contrast to the GAN methods in Tab. 6, which are ConvNet-based, our pMF adopts large-patch Vision Transformers, which are more FLOPs-efficient. For example, StyleGAN-XL (Sauer et al., 2022) costs 1574 Gflops per forward, 5.8× more than our pMF-H/16.

Compared to multi-step and/or latent-based methods, pMF remains competitive and substantially narrows the gap.

**ImageNet 512×512.** Tab. 7 shows that pMF achieves 2.48 FID at 512×512. Notably, it produces these results with a computational cost (in terms of both parameter count and Gflops) *comparable* to its 256×256 counterpart. In fact, the only overhead comes from the patch embedding and pre-

*Table 6.* **System-level comparison on ImageNet 256×256 generation.** FID and IS (Salimans et al., 2016) are evaluated on 50k samples, reported with CFG if applicable. ×2 in NFEs indicates that CFG doubles NFEs at inference time. All parameters and Gflops are reported as "generator (decoder)" for latent-space models. Gflops are for a single forward pass. The properties of 1-NFE or pixel-space are highlighted by blue. [1] Peebles & Xie 2023, [2] Ma et al. 2024, [3] Yu et al. 2025a, [4] Zheng et al. 2026, [5] Dhariwal & Nichol 2021, [6] Hoogeboom et al. 2023, [7] Kingma & Gao 2023, [8] Hoogeboom et al. 2025, [9] Li & He 2025, [10] Yu et al. 2025b, [11] Song et al. 2024, [12] Frans et al. 2025, [13] Geng et al. 2025a, [14] Geng et al. 2025b, [15] Brock et al. 2019, [16] Sauer et al. 2022, [17] Kang et al. 2023, [18] Lei et al. 2026.

| ImgNet 256×256 | NFE | space | params | Gflops | FID ↓ | IS ↑ |
|---|---|---|---|---|---|---|
| *Multi-step Latent-space Diffusion/Flow* | | | | | | |
| DiT-XL/2 [1] | 250×2 | latent | 675M (49M) | 119 (310) | 2.27 | 278.2 |
| SiT-XL/2 [2] | 250×2 | latent | 675M (49M) | 119 (310) | 2.06 | 277.5 |
| SiT-XL/2 + REPA [3] | 250×2 | latent | 675M (49M) | 119 (310) | 1.42 | 305.7 |
| RAE + DiT$^{DH}$-XL/2 [4] | 50×2 | latent | 839M (415M) | 146 (106) | 1.13 | 262.6 |
| *Multi-step Pixel-space Diffusion/Flow* | | | | | | |
| ADM-G [5] | 250×2 | pixel | 554M | 1120 | 4.59 | 186.7 |
| SiD, UViT [6] | 1000×2 | pixel | 2.5B | 555 | 2.44 | 256.3 |
| VDM++ [7] | 256×2 | pixel | 2.5B | 555 | 2.12 | 267.7 |
| SiD2, Flop Heavy [8] | 512×2 | pixel | - | 653 | 1.38 | - |
| JiT-G/16 [9] | 100×2 | pixel | 2B | 383 | 1.82 | 292.6 |
| PixelDiT-XL/16 [10] | 100×2 | pixel | 797M | 311 | 1.61 | 292.7 |
| *1-NFE Latent-space Diffusion/Flow* | | | | | | |
| iCT-XL/2 [11] | 1 | latent | 675M (49M) | 119 (310) | 34.24 | - |
| Shortcut-XL/2 [12] | 1 | latent | 676M (49M) | 119 (310) | 10.60 | 102.7 |
| MeanFlow-XL/2 [13] | 1 | latent | 676M (49M) | 119 (310) | 3.43 | 247.5 |
| iMF-XL/2 [14] | 1 | latent | 610M (49M) | 175 (310) | 1.72 | 282.0 |
| *1-NFE Pixel-space GAN* | | | | | | |
| BigGAN-deep [15] | 1 | pixel | 56M | 59 | 6.95 | 171.4 |
| StyleGAN-XL [16] | 1 | pixel | 166M | 1574 | 2.30 | 260.1 |
| GigaGAN [17] | 1 | pixel | 569M | - | 3.45 | 225.5 |
| *1-NFE Pixel-space Diffusion/Flow* | | | | | | |
| EPG-L/16 [18] | 1 | pixel | 540M | 113 | 8.82 | - |
| **pMF-B/16 (ours)** | 1 | pixel | 118M | 33 | 3.12 | 254.6 |
| **pMF-L/16 (ours)** | 1 | pixel | 410M | 117 | 2.52 | 262.6 |
| **pMF-H/16 (ours)** | 1 | pixel | 956M | 271 | **2.22** | 268.8 |

diction layers, which have more channels; all Transformer blocks maintain the same computational cost.

**Overhead of latent decoders.** We note that, with the progress of one-step methods, the overhead of the latent *decoder* is no longer negligible. This overhead has frequently been overlooked in prior studies. For example, the standard SD-VAE decoder (Rombach et al., 2022) takes 310G and 1230G flops at resolution 256 and 512, which alone exceeds the computational cost of our entire generator.

## 7. Conclusion

In essence, an image generation model is a mapping from noise to image pixels. Due to the inherent challenges of generative modeling, the problem is commonly decomposed into more tractable subproblems, involving multiple steps and stages. While effective, these designs deviate from the end-to-end spirit of deep learning.

Our study on pMF suggests that neural networks are highly expressive mappings and, when appropriately designed, are capable of learning complex end-to-end mappings, *e.g.*, directly from noise to pixels. Beyond its practical potential, we hope that our work will encourage future exploration of direct, end-to-end generative modeling.

*Table 7.* **System-level comparison on ImageNet 512×512 generation.** pMF employs an aggressive patch size of 32, resulting in low computational cost similar to 256×256, while achieving strong performance. Notations are similar to Tab. 6. [1] Peebles & Xie 2023, [2] Ma et al. 2024, [3] Yu et al. 2025a, [4] Zheng et al. 2026, [5] Dhariwal & Nichol 2021, [6] Hoogeboom et al. 2023, [7] Kingma & Gao 2023, [8] Hoogeboom et al. 2025, [9] Li & He 2025, [10] Lu & Song 2025, [11] Hu et al. 2025, [12] Brock et al. 2019, [13] Sauer et al. 2022.

| ImgNet 512×512 | NFE | space | params | Gflops | FID ↓ | IS ↑ |
|---|---|---|---|---|---|---|
| *Multi-step Latent-space Diffusion/Flow* | | | | | | |
| DiT-XL/2 [1] | 250×2 | latent | 675M (49M) | 525 (1230) | 3.04 | 240.8 |
| SiT-XL/2 [2] | 250×2 | latent | 675M (49M) | 525 (1230) | 2.62 | 252.2 |
| SiT-XL/2 + REPA [3] | 250×2 | latent | 675M (49M) | 525 (1230) | 2.08 | 274.6 |
| RAE + DiT$^{DH}$-XL/2 [4] | 50×2 | latent | 831M (415M) | 642 (408) | 1.13 | 259.6 |
| *Multi-step Pixel-space Diffusion/Flow* | | | | | | |
| ADM-G [5] | 250×2 | pixel | 559M | 1983 | 7.72 | 172.7 |
| SiD, UViT [6] | 1000×2 | pixel | 2.5B | 555 | 3.02 | 248.7 |
| VDM++ [7] | 256×2 | pixel | 2.5B | 555 | 2.65 | 278.1 |
| SiD2, Flop Heavy [8] | 512×2 | pixel | - | 653 | 1.48 | - |
| JiT-G/32 [9] | 100×2 | pixel | 2B | 384 | 1.78 | 306.8 |
| *1-NFE Latent-space Diffusion/Flow* | | | | | | |
| sCT-XXL [10] | 1 | latent | 1.5B (49M) | 552 (1230) | 4.29 | - |
| MeanFlow-RAE [11] | 1 | latent | 841M (415M) | 643 (408) | 3.23 | - |
| *1-NFE Pixel-space GAN* | | | | | | |
| BigGAN-deep [12] | 1 | pixel | 56M | 76 | 7.50 | 152.8 |
| StyleGAN-XL [13] | 1 | pixel | 168M | 2061 | 2.41 | 267.8 |
| *1-NFE Pixel-space Diffusion/Flow* | | | | | | |
| **pMF-B/32 (ours)** | 1 | pixel | 120M | 34 | 3.70 | 271.9 |
| **pMF-L/32 (ours)** | 1 | pixel | 413M | 117 | 2.75 | 276.8 |
| **pMF-H/32 (ours)** | 1 | pixel | 959M | 272 | 2.48 | 284.9 |

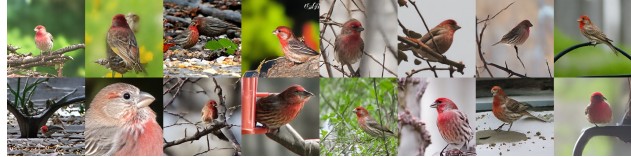

class 12: house finch, linnet, Carpodacus mexicanus

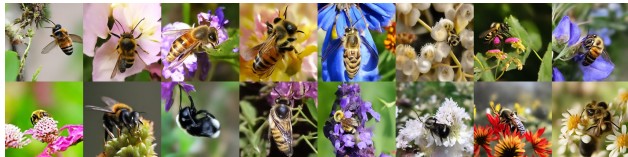

class 309: bee

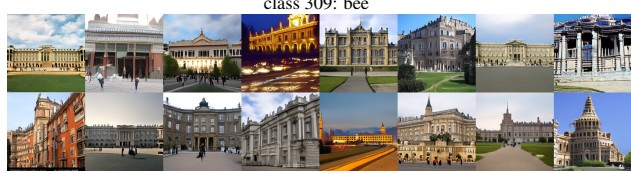

class 698: palace

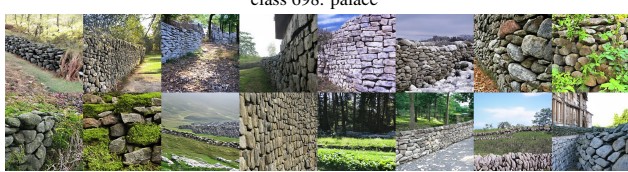

class 825: stone wall

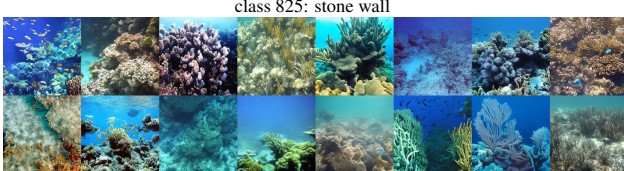

class 973: coral reef

*Figure 4.* **Qualitative results of 1-NFE pixel-space generation on ImageNet 256×256.** We show *uncurated* results of pMF-H/16 on the five classes listed here; more are in Appendix B.

## Acknowledgements

We greatly thank Google TPU Research Cloud (TRC) for granting us access to TPUs. S. Lu, Q. Sun, H. Zhao, Z. Jiang and X. Wang are supported by the MIT Undergraduate Research Opportunities Program (UROP). We thank our group members for helpful discussions and feedback.

## Impact Statement

This paper presents work whose goal is to advance the field of Machine Learning. There are many potential societal consequences of our work, none which we feel must be specifically highlighted here.

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

*Table 8.* **Configurations and hyper-parameters**. [†]: for ablation studies. [1] Li & He 2025, [2] Jordan et al. 2024, [3] Goyal et al. 2017.

| configs | pMF-B | pMF-L | pMF-H |
|---|---|---|---|
| depth | 16 | 32 | 48 |
| hidden dim | 768 | 1024 | 1280 |
| attn heads | 12 | 16 | 16 |
| patch size | | image_size / 16 | |
| noise scale | | image_size / 256 | |
| aux-head depth | | 8 | |
| class tokens | | 8 | |
| time tokens | | 4 | |
| guidance tokens | | 4 | |
| interval tokens | | 4 | |
| bottleneck dim [1] | 128 | 128 | 256 |
| linear layer init | | $\mathcal{N}(0, \sigma^2), \sigma^2 = 0.1/\texttt{fan\_in}$ | |
| epochs | $160^\dagger$ / 320 | 320 | 360 |
| batch size | | 1024 | |
| optimizer | | Muon [2], with $(\beta_1, \beta_2) = (0.9, 0.95)$ | |
| learning rate | | constant 1e-3 | |
| lr warmup [3] | | 0 epoch | |
| weight decay, dropout | | 0.0 | |
| ema half-life (Mimgs) | | {500, 1000, 2000} | |
| ratio of $r \neq t$ | | 50% | |
| $(t, r)$ cond | | $t - r$ | |
| $t, r$ sampler | | logit-normal(0.8, 0.8) | |
| cls drop [3] | | 0.1 | |
| CFG dist $\beta$ | 1 | 2 | 2 |
| LPIPS weight | | 0.4 | |
| ConvNeXt weight | | 0.1 | |
| threshold $t_{\text{thr}}$ | | 0.8 | |

## A. Implementation Details

### A.1. Configurations

The configurations and hyper-parameters are summarized in Tab. 8. Our implementation is based on iMF (Geng et al., 2025b), which is based on JAX and TPUs.

**CFG.** We strictly follow iMF's CFG implementation, with the network conditioned on CFG scale interval. The CFG scale and interval sampling strategy during training remains the same. FID results are evaluated at optimal guidance scale and interval. The pseudo-code[2] is provided in Alg. 2.

**EMA.** Our EMA implementation follows EDM (Karras et al., 2022). We maintain several EMA decay rates and select the best-performing one during inference.

**Perceptual Loss.** We use the standard LPIPS (Zhang et al., 2018) based on the VGG classifier and a variant based on ConvNeXt-V2 (Woo et al., 2023) as perceptual losses. Our implementation follows Lu et al. (2025). Additionally, we apply random crop and resize to 224×224 on both images (generated and ground-truth) before we apply perceptual loss, serving as an augmentation on segmentation signals.

---

[2]For brevity, we omit the implementation of guidance interval in the pseudo-code.

**Algorithm 2** pixel MeanFlow: training guidance.

Note: in PyTorch and JAX, jvp returns the function output and JVP.

```
# net: x-prediction network
# x, c: training and condition batch

t, r, w = sample_t_r_cfg()
e = randn_like(x)

z = (1 - t) * x + t * e

# average velocity u from x-prediction
def u_fn(z, r, t, w, c):
    return (z - net(z, r, t, w, c)) / t

# cond and uncond instantaneous velocity v
v_c = u_fn(z, t, t, w, c)
v_u = u_fn(z, t, t, w, None)

# Compute CFG target (same as iMF)
v_g = (e - x) + (1 - 1 / w) * (v_c - v_u)

# predict u and dudt
u, dudt = jvp(u_fn, (z, r, t, w, c),
                    (v_c, 0, 1, 0, 0))

# compound function V
V = u + (t - r) * stopgrad(dudt)

loss = metric(V, stopgrad(v_g))
```

**Longer training.** For Tabs. 6 and 7, we adopt a slightly modified training setup to better suit longer runs. Specifically, we double the noise scale by using a logit-normal time sampler logit-normal(0.0, 0.8). In addition, to obtain a smoother sampling distribution, we sample $(t, r)$ uniformly from $[0, 1]$ with 10% probability (instead of always using the default sampler). Finally, we reduce the threshold $t_{\text{thr}}$ to 0.6 to account for the increased noise scale.

### A.2. Visualization of the generalized denoised images

In Fig. 1, we visualize the underlying *average velocity field* **u** and the induced *generalized denoised images* **x** by simulating an ODE trajectory from $t = 1$ to $t = 0$. The images of **u** are shown as $-\mathbf{u}$ for better visualization. We use the pretrained JiT-H/16 to obtain the instantaneous velocity **v** and solve the ODE trajectory $\{\mathbf{z}_t\}_{t=0}^1$ via a numerical ODE solver. Based on the simulated trajectory, we compute **u** and **x** for different $(r, t)$ pairs via Eq. (5) and Eq. (8).

## B. Visualizations

We provide additional qualitative results in Fig 5 and Fig 6. These results are uncurated samples of the classes listed as conditions. These results are from our pMF-H/16 model for 1-NFE ImageNet 256×256 generation. Here, we set CFG scale $\omega = 7.0$ and CFG interval $[0.1, 0.7]$. This evaluation setting corresponds to an FID of 2.74 and an IS of 290.0.

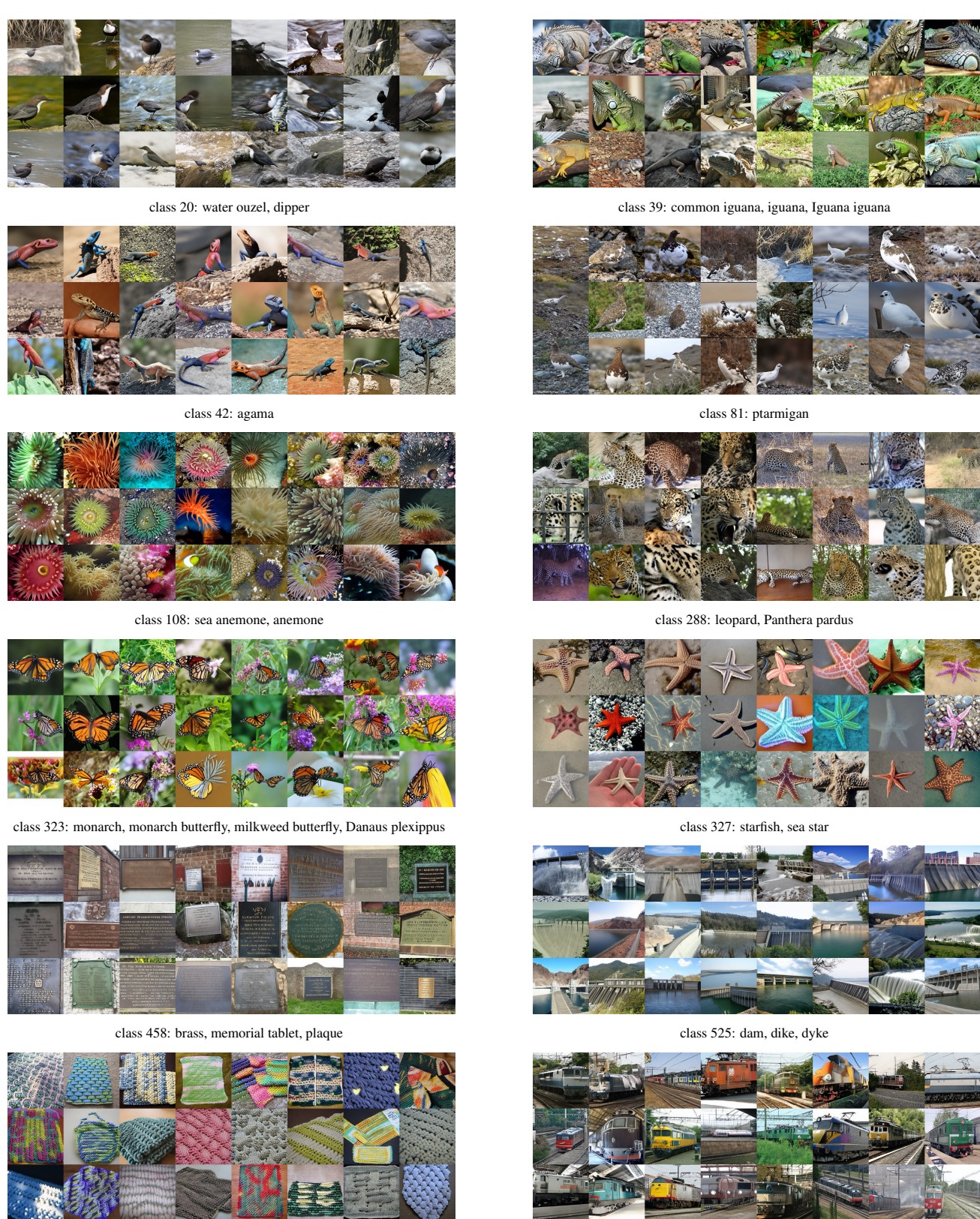

class 20: water ouzel, dipper

class 39: common iguana, iguana, Iguana iguana

class 42: agama

class 81: ptarmigan

class 108: sea anemone, anemone

class 288: leopard, Panthera pardus

class 323: monarch, monarch butterfly, milkweed butterfly, Danaus plexippus

class 327: starfish, sea star

class 458: brass, memorial tablet, plaque

class 525: dam, dike, dyke

class 533: dishrag, dishcloth

class 547: electric locomotive

*Figure 5. Uncurated* 1-NFE pixel class-conditional generation samples of pMF-H/16 on ImageNet 256×256.

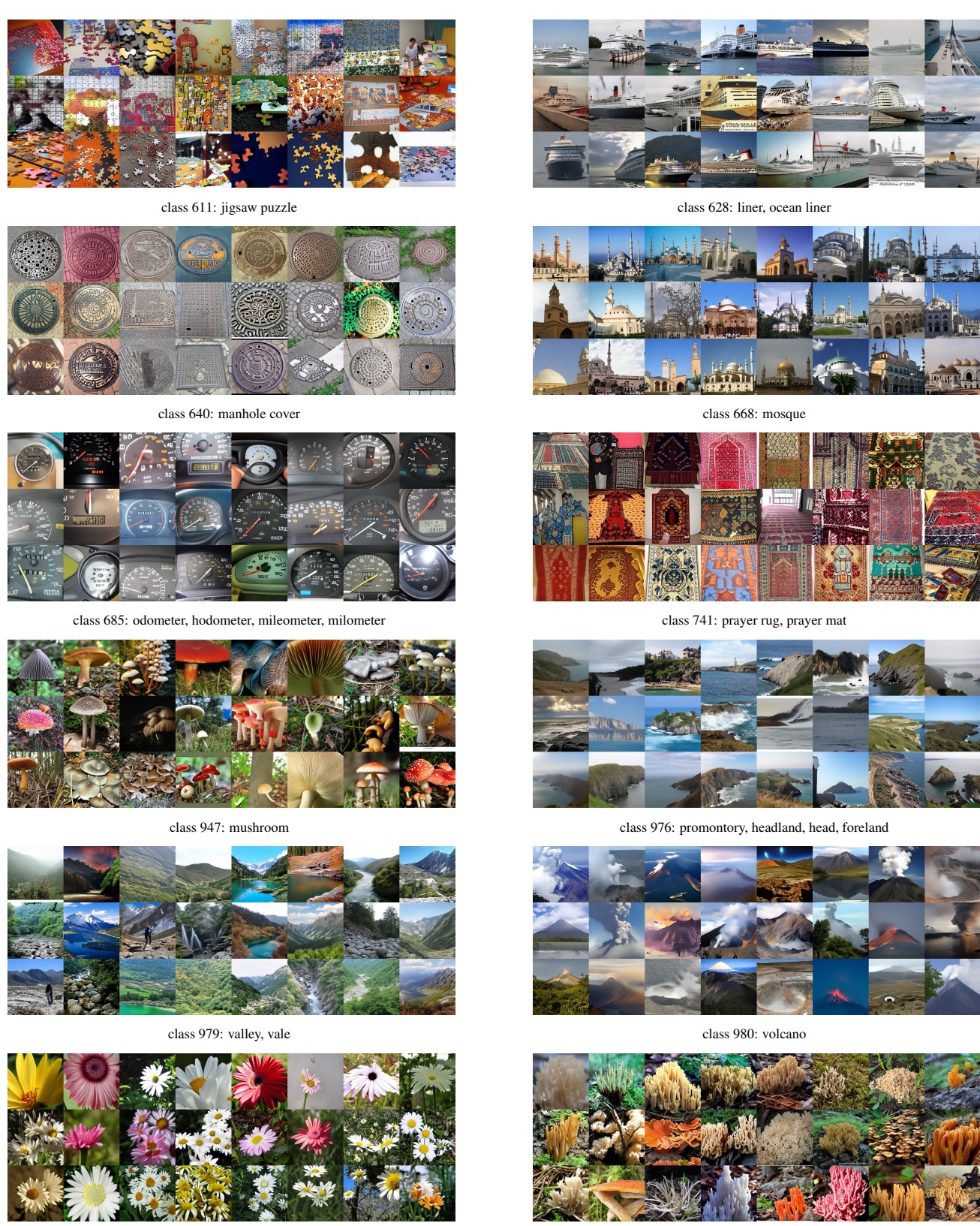

class 611: jigsaw puzzle

class 628: liner, ocean liner

class 640: manhole cover

class 668: mosque

class 685: odometer, hodometer, mileometer, milometer

class 741: prayer rug, prayer mat

class 947: mushroom

class 976: promontory, headland, head, foreland

class 979: valley, vale

class 980: volcano

class 985: daisy

class 991: coral fungus

*Figure 6. Uncurated* 1-NFE pixel class-conditional generation samples of pMF-H/16 on ImageNet 256×256.

