# OpenReview forum: "One-step Latent-free Image Generation with Pixel Mean Flows"
_ICML.cc/2026/Conference — ICML 2026 regular_

### Official Review · Reviewer_q3iA · 2026-02-15

**Soundness:** 4
**Presentation:** 4
**Significance:** 3
**Originality:** 3
**Overall Recommendation:** 5
**Confidence:** 4

**Summary:**

The paper proposes **pixel MeanFlow (pMF)**, a one-step image generation approach that operates **directly in pixel space** without relying on a pretrained latent tokenizer/decoder. The core idea is to let the network **predict denoised images (x-prediction)**—which is more learnable in high-dimensional pixel representations—while training it using **MeanFlow/iMF-style velocity-space supervision** through a simple conversion from the predicted denoised image to the corresponding flow/velocity target. This unifies the practicality of pixel-space generation with the modeling benefits of MeanFlow objectives, and it naturally enables adding pixel-domain perceptual losses.

Empirically, the authors show that directly predicting velocity/flow in pixel space becomes unstable at higher resolutions, whereas x-prediction remains trainable, motivating the design. They report strong one-step, latent-free results on class-conditional ImageNet (including 256×256 and 512×512) and provide ablations on time sampling, parameterization, optimizer, and perceptual loss that explain which components are necessary for performance.

**Compliance With Llm Reviewing Policy:**

Affirmed.

**Final Justification:**

I appreciate the authors’ detailed and thoughtful rebuttal, which has fully addressed all my questions. I have no further concerns and will maintain my existing review recommendation. My initial assessment was intentionally positive, and already reflects my view that the revised manuscript falls into the borderline accept range.

**Key Questions For Authors:**

## Key Questions for Authors

1. **Role of bottleneck embedding vs. VAE-like compression**
   - You mention a *bottleneck embedding* (and note it is excluded in some ablations). Could you clarify its exact architecture and purpose (e.g., input dimension → bottleneck dimension → model width)?
   - To what extent does this bottleneck act as an **implicit compression module** analogous to a VAE tokenizer/encoder (but deterministic), and how does it affect the model’s information capacity?

2. **Potential loss of high-frequency details**
   - Does the bottleneck embedding introduce measurable degradation in **high-frequency content** (textures, edges), especially at higher resolutions / larger patch sizes?
   - Have you quantified this with frequency-aware metrics or analyses (e.g., spectral error, edge statistics), or by controlled comparisons with/without the bottleneck at fixed compute? A short study would help assess whether the bottleneck trades detail for stability.

3. **Is the model “distilling” an implicit autoencoder during training?**
   - Since the network outputs pixel-space denoised images while being trained via a velocity-space objective, do you view the bottleneck + transformer stack as effectively learning an **implicit autoencoding/tokenization** structure during training?
   - If so, how does this implicit representation compare to explicit VAE tokenizers (e.g., in reconstruction fidelity or frequency response), and is there evidence that the model learns a “latent-like” manifold internally?

4. **Training cost trade-offs introduced by MeanFlow/iMF machinery (r,t sampling and JVP)**
   - Your approach requires sampling over the full \((r,t)\) region and computing JVPs (Algorithm 1). Can you report the **training-time overhead** relative to (a) standard Flow Matching and (b) one-step latent approaches (including VAE training / tokenizer pretraining)?
   - Concretely: what is the wall-clock slowdown, additional memory, and compute (FLOPs) cost attributable to JVP and the 2D time sampling?

5. **Is the added training complexity necessary for the reported gains?**
   - Beyond the ablations that show restricted time sampling fails, can you provide a clearer *cost–benefit* breakdown: how much quality improvement is attributable to (i) full \((r,t)\) coverage, (ii) JVP term, (iii) perceptual loss, and (iv) optimizer choice?
   - This would help readers understand whether the main benefit comes from the MeanFlow/iMF formulation itself or from the overall training recipe.

6. **Practical implications when guidance is used**
   - Since guidance can increase NFEs at inference, do the training-cost increases (JVP, full \((r,t)\)) still pay off in practical guided settings? Any results comparing unguided vs guided regimes would be helpful for deployment-oriented readers.

**Limitations:**

No.

The paper includes an Impact Statement, but it does not meaningfully discuss limitations or potential negative societal impact (it states that although there may be many societal consequences, none need to be specifically highlighted). I suggest the authors expand this section in a concrete and paper-specific way, including:

- **Misuse and harm**: risks typical of high-fidelity image generation (e.g., deepfakes/misinformation, impersonation, privacy violations, harmful/illegal content), and practical mitigation steps (data governance, content filtering, provenance/watermarking, responsible release).
- **Methodological limitations**: explicitly highlight the lack of guarantee for the “denoised-image-like / manifold” assumption in the general case \(0<r<t\) (Sec. 4.2), and clarify where the method may fail.
- **Compute/training trade-offs**: quantify training overhead introduced by full \((r,t)\) coverage and JVP/stop-gradient machinery (Alg. 1 / Sec. 4.3), and compare training cost to latent-tokenizer (VAE) pipelines and simpler one-step objectives.
- **Practical deployment caveats**: note that inference NFEs can increase under guidance (as indicated in the system-level comparison tables), and discuss how this affects the “one-step” benefits in guided settings.

**Strengths And Weaknesses:**

## Strengths

- **Strong results in the hardest regime (one-step + latent-free + pixel-space).**
  The paper directly targets *1-NFE* generation without a latent tokenizer/decoder and operates in pixel space. It reports very strong ImageNet results (see **Abstract**; also supported by the system-level comparisons in **Sec. 6**, **Tab. 6–7**).

- **Clean, unified formulation that separates “prediction space” from “loss space” while keeping inference simple.**
  The key design is to **predict denoised images (x-prediction)** while **training with MeanFlow/iMF-style velocity-space supervision** through a simple conversion \(x \rightarrow u \rightarrow V\). This decoupling is explicitly summarized in **Tab. 1** and implemented in **Sec. 4.3 / Alg. 1**.

- **Convincing ablations that justify the design choices.**
  The paper pinpoints why **x-prediction is crucial** for high-dimensional pixel modeling (the u-pred alternative collapses at higher resolution; **Tab. 2**) and shows that several alternative designs underperform the final recipe (**Tab. 3**). This makes the empirical story tight rather than “just another recipe.”


## Weaknesses

- **Weaker theoretical guarantees for the central manifold-style assumption.**
  The paper explicitly notes that for the general case \(0<r<t\), \(x(z_t,r,t)\) is **not guaranteed** to correspond to a data-manifold image (only motivated empirically and by simulations). See **Sec. 4.2** discussion around the “general case”.

- **Training complexity / efficiency concerns (JVP + stop-gradient), potentially less straightforward than VAE-latent pipelines.**
  Although inference avoids a latent decoder, training relies on **Jacobian-vector products (JVP)** and stop-gradient construction in **Alg. 1 (Sec. 4.3)**. This adds engineering complexity and may introduce nontrivial training overhead; the paper emphasizes inference-system benefits but provides limited quantification of this training-side cost.

- **High sensitivity to optimizer and loss recipe.**
  The results depend strongly on specific choices such as **Muon vs Adam** and adding **perceptual losses**, which produce large FID changes (see **Fig. 3a–b**). This suggests reproducibility/portability may require careful tuning.

- **Guidance / broader conditioning is not fully demonstrated in the main evaluation.**
  CFG support is mentioned (with details deferred), and the paper notes that **CFG doubles NFEs** in the system tables (**Tab. 6–7**). Moreover, the main experiments are class-conditional ImageNet; text-conditional generation is not a central evaluated setting in the core results (**Sec. 6**).

---

> ### Author Rebuttal · Authors · 2026-03-31
>
> We sincerely thank the reviewer for the positive assessment of our paper, especially for recognizing the strong results in the challenging one-step, latent-free, pixel-space regime, the clean prediction-space / loss-space decoupling, and the comprehensive ablations. We also appreciate the thoughtful questions on the bottleneck embedding, training cost, guidance, and limitations. We address them below.
>
> **(1) Role of the bottleneck embedding and connection to compression learning.**
>
> We thank the reviewer for raising this thoughtful question. The bottleneck embedding is a lightweight input projection before the main Transformer, implemented as a 2-layer MLP: input/patch dimension → bottleneck dimension → model width. Its main purpose is to ease the transition from raw high-dimensional pixel inputs into the backbone model.
>
> We do not view it as analogous to a VAE-style tokenizer or compression module. The key difference is that the bottleneck sits only at the very front of the network, with highly asymmetric capacity on the two sides: the input side is still a shallow local image embedding, while the main representational power remains in the backbone Transformer. In this sense, it may relate to manifold learning or compact input representation (as also discussed in JiT [1]), but it is not intended as a separate learned latent space and is not the main focus of our method. We will clarify this distinction in the revision.
>
> **(2) On potential loss of high-frequency details from the bottleneck embedding.**
>
> Based on our current visual inspection of generated samples, we do not observe an obvious degradation of high-frequency details such as textures or edges due to the bottleneck embedding.
>
> Analyses that are more directly targeted at high-frequency content, such as frequency-based or edge-based metrics, would help make this tradeoff much clearer. We will add this discussion in the revision and note this as a useful direction for future evaluation.
>
> **(3) Training cost relative to Flow Matching and one-step latent approaches.**
>
> We appreciate the reviewer’s question on the training cost. Compared with standard pixel-space Flow Matching (FM) and one-step latent approaches, our estimated training compute is as follows:
>
> - pixel-space FM: 0.86e20 FLOPs
> - pMF: 1.55e20 FLOPs
> - iMF: 1.17e20 FLOPs
> - tokenizer: 2.39e20 FLOPs
>
> One-step latent approaches typically rely on separately pre-trained tokenizers, whose training and deployment costs are often not reflected in the reported generator-only FLOPs. Following the reviewer’s suggestion, we report the tokenizer training cost as a reference. Under this accounting, the total training cost of pMF is lower than the combined cost of training a one-step latent space generative models with its tokenizer.
>
> We also provide the breakdown of JVP and 2D time sampling in the following section. We agree that presenting this analysis more clearly would strengthen the paper, and we will revise the paper accordingly.
>
> **(4) Gains and costs from each technique.**
>
> We thank the reviewer for this thoughtful question on the gains and costs of the individual techniques. Our current ablations provide the gains for individual techniques.
>
> - Full 2D time sampler over $(r,t)$ is important for performance: as shown in Table 3(b), restricting time sampling to only $r=t$, only $r=0$, or both lines causes severe failure relative to the full $(r,t)$ coverage. This choice does not introduce additional training cost in practice.
> - The JVP term is a necessary component in MeanFlow/iMF formulation, and is central to enabling one-step generation. Its overhead is about 35% additional training FLOPs and 20% wall-clock time, with no noticeable memory increase in our compiled implementation.
> - Optimizer choice also matters: Muon improves both convergence speed and final performance, as shown in Figure 3(a), with almost no additional FLOPs, about 17% wall-clock overhead.
> - Perceptual loss is not required for pMF to function, but it further improves FID, as shown in Figure 3(b). Its overhead is about 33% additional training FLOPs and 34% wall-clock time.
>
> We agree that presenting this cost–benefit breakdown more explicitly would make the paper stronger, and we will add a clearer discussion in the revision.
>
> **(5) Practical implications when guidance is used.**
>
> Thank you for raising this point. We would like to clarify that the guidance mechanism used in our implementation is not the standard classifier-free guidance (CFG) that requires two forward passes at inference time. Instead, following the iMF implementation, we treat the guidance scale/interval as an additional conditioning input to the network, so guidance is incorporated into training, while inference only requires one single forward pass. We will clarify this more explicitly in the revision to avoid confusion.
>
> [1] Li, T. and He, K. Back to basics: Let denoising generative models denoise. arXiv preprint arXiv:2511.13720, 2025.

---

> > ### Author Rebuttal · Reviewer_q3iA · 2026-04-02
> >
> > Thank you for the detailed rebuttal. My main concerns are partially resolved.
> >
> > The rebuttal usefully clarifies several important points. In particular, the authors now explain the architectural role of the bottleneck embedding more clearly, provide a helpful compute breakdown relative to pixel-space FM / iMF / tokenizer-based pipelines, and clarify that their guidance implementation does not require the standard two-forward-pass CFG at inference. The additional discussion of the roles of full $(r,t)$ sampling, JVP, optimizer choice, and perceptual loss also makes the method and training recipe easier to understand.
> >
> > That said, two parts of my original review are only partially addressed.
> >
> > First, regarding the bottleneck embedding and possible loss of high-frequency details, the rebuttal mainly states that no obvious degradation is observed in visual inspection, and that more targeted frequency- or edge-based analysis would be useful. I appreciate this clarification, but this does not yet fully answer the question of whether the bottleneck trades detail for stability, especially at higher resolutions.
> >
> > Second, on whether the bottleneck + transformer stack is implicitly learning a latent-like or autoencoding structure, the rebuttal clarifies that the authors do not intend it to function like a VAE tokenizer. This is helpful, but the response remains mostly conceptual. I still think the paper would benefit from a more explicit discussion of what this bottleneck is and is not doing representationally, even if a full empirical study is deferred to the revision.
> >
> > Overall, I think the rebuttal addresses the main concerns constructively and improves the clarity of the paper. My remaining concerns are relatively limited and mostly about the strength of evidence for these interpretive questions, rather than about the core empirical validity of the method.

---

### Official Review · Reviewer_Jyse · 2026-03-12

**Soundness:** 4
**Presentation:** 4
**Significance:** 3
**Originality:** 3
**Overall Recommendation:** 4
**Confidence:** 4

**Summary:**

This paper presents a pixel-space meanflow model that achieves superior performance for one-step image generation. To achieve this, the authors design the network target as an image on a low-dimensional image manifold and propagate the loss computed in the velocity space. By operating in pixel space, image-level perceptual loss can be applied to enhance generation quality.

**Compliance With Llm Reviewing Policy:**

Affirmed.

**Final Justification:**

I have no further questions for the rebuttal and will maintain my positive rating. For the final revision, I look forward to seeing the results of the pixel-space rectified-flow and reflow experiments, ideally utilizing the same model architecture and training pipeline to ensure a fair comparison.

**Key Questions For Authors:**

1. One motivation for training diffusion models in the latent space is to accelerate scalable training. Will pixel-level MeanFlow training converge more slowly in larger-scale settings (e.g., text-to-image generation)?

2. Applying perceptual loss is one of the motivations for pixel-space MeanFlow. However, it is unclear whether this auxiliary loss (for image reconstruction) may break the marginal preservation of the velocity field required by flow models for theoretical soundness. Additionally, could the perceptual loss encourage memorization rather than generalization?

3. As mentioned in the paper, the authors employ advanced components such as the Muon optimizer and appear to carefully design the model architecture and training pipeline. Could these engineering choices be a major factor contributing to the improved performance? A contrast experiment comparing a baseline (e.g., rectified flow + reflow) in pixel space could help demonstrate the effectiveness of the proposed method.

**Limitations:**

Some of the points raised in the "Key Questions" section may also be considered potential limitations of the work.

**Strengths And Weaknesses:**

**Strengths**

+ The experiments in this paper are comprehensive and include essential ablation studies.
+ The proposed method outperforms the compared methods by a remarkable margin for one-step image generation.
+ The authors appear to have invested significant engineering effort in designing and training the models.

**Weaknesses**

- Limited methodological novelty. At the method level, the proposed approach appears to be a combination of JiT and MeanFlow. The discussed x-prediction has been explored in previous works such as EDM and JiT, while the one-step generation capability seems mainly attributed to MeanFlow.
- It is unclear whether training efficiency is reduced due to pixel-level flow model training.
- It is unclear whether the use of perceptual loss may lead to more severe memorization issues. It would be helpful to report metrics such as precision/recall to better measure the diversity of generated samples relative to the training data.

---

> ### Author Rebuttal · Authors · 2026-03-31
>
> We sincerely thank the reviewer for the positive assessment of our paper, especially for recognizing the comprehensive ablations, the strong empirical performance in one-step generation, and the substantial effort in model/training design. We also appreciate the thoughtful questions on novelty, scalability/training efficiency, perceptual loss, memorization, and the role of engineering choices.
>
> **(1) Will pixel-level MeanFlow training converge more slowly in larger-scale settings (e.g., text-to-image generation)?**
>
> We thank the reviewer for raising this important question. In our current ImageNet experiments, we do not observe slower convergence for pixel-level MeanFlow training itself. In fact, when equipped with perceptual loss, pMF converges even faster than its latent counterpart in our comparisons.
>
> Our main motivation for studying pixel-space generation is therefore different from simply pursuing faster optimization. Rather, we aim to remove the need for separately pretrained tokenizers/decoders, which can constitute a substantial additional cost in latent-space pipelines, and to enable fully feedforward generative modeling directly in pixel space. We will clarify this distinction more explicitly in the revision.
>
> At the same time, we agree that larger-scale settings such as text-to-image generation are an important next step. We don’t observe a clear convergence speed difference in our preliminary experiments. Our current submission focuses on class-conditional ImageNet as a controlled setting for isolating the core formulation, and we view broader multimodal scaling as an important direction for future work.
>
> **(2) On whether perceptual loss may encourage memorization or break the MeanFlow trajectory.**
>
> We thank the reviewer for raising this point. We agree that perceptual loss introduces a meaningful tradeoff and should be evaluated not only by sample quality but also by diversity and theoretical consistency.
>
> Empirically, we do not observe evidence that perceptual loss simply increases memorization at the expense of diversity. On ImageNet, pMF-B/16 trained with only MSE achieves precision = 74.8 and recall = 39.5, while our pMF-B/16 with perceptual loss achieves precision = 81.3 and recall = 50.0. This suggests that perceptual loss improves not only fidelity (precision) but also coverage (recall), indicating better overall generation quality rather than a trivial shift toward memorization.
>
> Regarding theoretical soundness, we agree that adding perceptual loss can, in principle, perturb the underlying MeanFlow trajectory. In our design, however, we apply perceptual loss only at low noise levels ($t < 0.8$), where where the image content is already relatively recoverable. In practice, we find that this localized use mainly serves to accelerate convergence and improve generation quality, while having a limited impact on the overall flow trajectory.
>
> In summary, our results suggest that it improves both fidelity and diversity in practice, and its effect on the underlying trajectory is mitigated by restricting it to the low-noise regime.
>
> **(3) On whether the gains mainly come from engineering choices, such as Muon, or architectural design.**
>
> We thank the reviewer for this important point and agree that engineering choices should be disentangled from the core method.
>
> For the optimizer, on JiT-B/16 trained for 200 epochs, Adam achieves FID 31.05 while Muon achieves 30.54 (both without CFG), suggesting that Muon alone provides only a modest improvement for a standard pixel-space baseline. Our interpretation is that Muon may be particularly helpful for MeanFlow-style training, where stronger early learning can compound more effectively, but it does not by itself explain the overall gains.
>
> For the architecture, we largely follow the iMF design and do not introduce major architectural modifications. We agree that a pixel-space rectified-flow / reflow baseline would be valuable for further isolating the method contribution, and we will incorporate this point in the revision.

---

> > ### Author Rebuttal · Reviewer_Jyse · 2026-04-04
> >
> > I have no further questions and will maintain my positive rating. For the final revision, I look forward to seeing the results of the pixel-space rectified-flow and reflow experiments, ideally utilizing the same model architecture and training pipeline to ensure a fair comparison.

---

### Official Review · Reviewer_HmCS · 2026-03-13

**Soundness:** 3
**Presentation:** 4
**Significance:** 3
**Originality:** 3
**Overall Recommendation:** 4
**Confidence:** 4

**Summary:**

This paper proposes pixel MeanFlow (pMF) for one-step image generation directly in pixel space without latent representations. The key idea is to decouple the prediction space and loss space: the network predicts a denoised image-like variable $x$, which is hypothesized to lie on a lower-dimensional image manifold, while training is performed using a velocity-space MeanFlow objective through a conversion
$x \rightarrow u \rightarrow v$. This formulation aims to make one-step generation in high-dimensional pixel space easier to learn. Experiments on ImageNet show strong results for one-step latent-free generation, achieving 2.22 FID at 256×256 and 2.48 FID at 512×512 with a single function evaluation. The paper also provides extensive ablations on prediction targets, optimizers, perceptual losses, and time samplers.

**Compliance With Llm Reviewing Policy:**

Affirmed.

**Final Justification:**

The paper is technically sound and clearly presented, and the additional clarifications help strengthen the overall understanding of the method. However, the contribution appears somewhat incremental, and the evaluation scope—focused on ImageNet—leaves open questions about generalization to more complex conditional settings such as text-to-image. Overall, my assessment remains unchanged, and I maintain my original score.

**Key Questions For Authors:**

- Can the authors provide stronger theoretical intuition or analysis explaining why the generalized $x(z_t,r,t)$ field should remain close to the image manifold in the general case?
- What is the final performance when training without perceptual loss but for longer training schedules (e.g., >1200 epochs)?
- Is the proposed formulation compatible with text-to-image generation, and if so, how does it perform compared to existing one-step or few-step methods?

**Limitations:**

yes

**Strengths And Weaknesses:**

Strengths
+ Conceptually simple formulation. The separation between prediction space and loss space provides a clear way to combine image prediction with MeanFlow training objectives.
+ Strong empirical performance. The method achieves competitive results among one-step pixel-space generative models and significantly improves over previous methods in the same setting.
+ Well-designed ablations. The paper includes thorough ablation studies on prediction targets, perceptual loss, optimizers, and time sampling strategies, helping justify design choices.

Weaknesses
- Limited theoretical justification. The core assumption that the generalized denoised field $x(z_t,r,t)$ lies close to a low-dimensional manifold is mainly supported empirically. In the general case $0<r<t$, the paper explicitly acknowledges that this property does not hold theoretically and relies primarily on intuition and visualizations.
- Unclear effect of perceptual loss. The perceptual loss contributes a large portion of the improvement (e.g., FID $9.56 \rightarrow 3.53$), and relies on networks pretrained on ImageNet. Since the baseline without perceptual loss is trained for relatively few epochs, it remains unclear whether the perceptual loss mainly accelerates convergence or improves the final performance ceiling. Reporting longer-training results without perceptual loss would help clarify this point.
- Limited evaluation scope. All experiments focus on class-conditional ImageNet generation. It would strengthen the paper to also include text-to-image experiments, which are more representative of modern diffusion model applications.

---

> ### Author Rebuttal · Authors · 2026-03-31
>
> We sincerely thank the reviewer for the careful reading and constructive feedback. We especially appreciate the reviewer’s recognition that the formulation is conceptually simple, that the empirical performance is strong, and that the ablations are well designed. We also thank the reviewer for the thoughtful questions regarding the theoretical intuition for the generalized denoised field, the role of perceptual loss, and the evaluation scope.
>
> **(1) On the theoretical intuition for why** $x(z_t,r,t)$ **should remain close to an image manifold in the general case** $0<r<t$**.**
>
> We agree that this paper does not provide a formal proof that the generalized denoised field always lies near a low-dimensional manifold in the most general setting. Our intention is to present this as a structural hypothesis rather than a theorem-level claim.
>
> This hypothesis is motivated by prior work [1], which suggests that denoised-image-like targets are intrinsically easier to model due to their lower-dimensional structure. In our case, the generalized denoised field behaves similarly to denoised images, which provides the underlying intuition.
>
> Importantly, this intuition is also supported by our empirical results. Both the toy examples and the ImageNet ablations consistently show that modeling this target is substantially easier and more effective than modeling the noisier alternative. We believe it is a well-motivated structural hypothesis with consistent empirical support.
>
> **(2) On the effect of perceptual loss.**
>
> Thank you for raising this point. Our current evidence suggests that perceptual loss improves both convergence speed and final performance. In particular, it substantially accelerates optimization: pMF-L/16 reaches 3.09 FID at 640 epochs without perceptual loss, versus 2.52 FID at 320 epochs with perceptual loss. A similar trend also holds under longer training schedules: without perceptual loss, pMF-B/16 reaches 4.94 FID at 1000 epochs, compared with 3.12 FID at 320 epochs with perceptual loss.
>
> Importantly, perceptual loss is not what makes the method work: even without it, the proposed $x$-prediction formulation already substantially outperforms $u$-prediction and related alternatives. We will include these additional results in the revision.
>
> **(3) On the limited evaluation scope and the question of text-to-image generation.**
>
> We thank the reviewer for this valuable suggestion and fully agree that text-to-image would be an important extension, especially as it is more representative of many modern diffusion model applications. In this work, we intentionally focus on class-conditional ImageNet to study the core method in a cleaner setting, and we will explicitly highlight broader multimodal scaling and text-to-image generation as important future directions.
>
> [1] Li, T. and He, K. Back to basics: Let denoising generative models denoise. arXiv preprint arXiv:2511.13720, 2025.

---

> > ### Author Rebuttal · Reviewer_HmCS · 2026-04-04
> >
> > Thank you for the detailed and thoughtful rebuttal. I appreciate the clarifications regarding the theoretical intuition and the additional results on the role of perceptual loss, which help address my concerns.
> >
> > Regarding the evaluation scope, I understand the choice of focusing on a cleaner ImageNet setting. However, since the method is presented as a general approach for one-step latent-free generation, it remains somewhat unclear how well it extends to more complex conditional scenarios (e.g., text-to-image). A brief discussion in this direction would further strengthen the paper.
> >
> > Overall, I find the responses satisfactory and will maintain my original score.

---

### Official Review · Reviewer_SAtk · 2026-03-13

**Soundness:** 4
**Presentation:** 3
**Significance:** 3
**Originality:** 4
**Overall Recommendation:** 5
**Confidence:** 3

**Summary:**

This paper proposes pixel Mean Flow (pMF), aiming to enable direct pixel-space image generation in a one-step and latent-free setting. Its core idea is to decouple the network output space from the loss space: the network directly predicts a denoised image-like quantity
x, which is closer to the image manifold, and then converts it through x→u→v so that training can still be supervised with the velocity-space loss used in Mean Flow. The authors argue that, compared with directly predicting the noisier velocity quantities u or v, predicting
x is easier to learn and better suited for high-dimensional pixel-space models.

The experimental results show that this design is clearly superior to direct u-prediction in high-dimensional pixel space, and that incorporating perceptual loss further improves performance. On ImageNet, pMF achieves strong 1-NFE generation results at both 256×256 and 512×512 resolutions, suggesting that one-step, latent-free, direct generation from noise to pixels is now both feasible and competitive.

**Compliance With Llm Reviewing Policy:**

Affirmed.

**Final Justification:**

After considering the overall discussion and the authors’ rebuttal, I view this paper as an interesting piece of work with a certain degree of theoretical justification. In particular, the proposed perspective offers a meaningful new angle for direct one-step generation in pixel space. While some theoretical questions remain open, I believe the paper introduces a fresh and thought-provoking direction for this problem setting. Overall, my assessment remains positive, and I keep my original score unchanged.

**Key Questions For Authors:**

see weaknesses

**Strengths And Weaknesses:**

Strengths
1. The method is strongly motivated, and its overall logic is clear. The authors identify a central challenge: one-step modeling benefits from the flow/velocity framework, but targets in the velocity space are too noisy to be learned effectively in pixel space. Therefore, they instead let the network predict x, which is closer to the image manifold. This motivation is reasonable.

2. The paper also provides a fairly clear explanation of why x-prediction is more suitable for pixel-space modeling. Through the idea of a generalized manifold hypothesis, the authors argue that, compared with the noisier quantities u and v, x is more like a denoised image and is therefore easier for the network to learn. This explanation is consistent with the experimental results. The method also shows clear advantages in high-dimensional pixel space, and the ablation studies are fairly comprehensive.

3. In addition, the design that decouples the prediction space from the loss space is quite novel. Rather than simply modifying the loss function or changing the backbone, the paper proposes that what the network outputs does not have to be the same as the space in which supervision is applied. This idea of “producing an image-like output while supervising it in the velocity space” is both new and insightful.The method is strongly motivated, and its overall logic is clear. The authors identify a central challenge: one-step modeling benefits from the flow/velocity framework, but targets in the velocity space are too noisy to be learned effectively in pixel space. Therefore, they instead let the network predict x, which is closer to the image manifold. This motivation is reasonable.

4. The paper also provides a fairly clear explanation of why x-prediction is more suitable for pixel-space modeling. Through the idea of a generalized manifold hypothesis, the authors argue that, compared with the noisier quantities u and v, x is more like a denoised image and is therefore easier for the network to learn. This explanation is consistent with the experimental results. The method also shows clear advantages in high-dimensional pixel space, and the ablation studies are fairly comprehensive.

5. In addition, the design that decouples the prediction space from the loss space is quite novel. Rather than simply modifying the loss function or changing the backbone, the paper proposes that what the network outputs does not have to be the same as the space in which supervision is applied. This idea of “producing an image-like output while supervising it in the velocity space” is both new and insightful.

weaknesses
1. The theoretical support is still somewhat limited, and the paper relies more on empirical intuition than on rigorous analysis. Although the paper emphasizes the generalized manifold hypothesis, this claim is mainly supported by intuition and visualization rather than by a strong theoretical proof. In other words, why
𝑥(𝑧𝑡,𝑟,𝑡) should generally be easier to learn is not yet theoretically well justified.

2. The authors define x(zt,r,t)=zt−t⋅u(zt,r,t),and then argue that this quantity behaves like a denoised image. Although this definition works well in practice, it is still somewhat hand-crafted, and its theoretical necessity is not fully established.

3. In addition, pMF is built on top of the Mean Flow / iMF formulation rather than being a completely independent framework. As a result, its applicability is still tied, to some extent, to this family of methods.

4. The paper also shows that the choice of pre-conditioner has a significant impact on performance. However, the analysis of why different pre-conditioners lead to such large performance differences remains mostly empirical and phenomenological, rather than providing a deeper explanation.

---

> ### Author Rebuttal · Authors · 2026-03-31
>
> We sincerely thank the reviewer for the positive assessment of our paper, and especially for recognizing the clarity of the motivation, the novelty of decoupling prediction space from loss space, and the strong empirical results in the challenging one-step, latent-free, pixel-space regime. We also appreciate the reviewer’s thoughtful questions regarding the theoretical justification, the definition of $x(z_t,r,t)$, the dependence on the MeanFlow/iMF formulation, and the role of pre-conditioning. We address them below.
>
> **(1) On the theoretical support for why** $x(z_t,r,t)$ **is easier to learn.**
>
> We agree that our paper does not provide a formal theorem establishing the superiority of x-prediction in full generality. Our intention is instead to offer a structural motivation, supported by both prior evidence and empirical validation.
>
> First, JiT suggests that clean images are intrinsically easier for neural networks to model because they lie on a substantially lower-dimensional manifold than noisy observations. Our target $x(z_t, r, t)$, as illustrated in Figure 1, can be interpreted as a generalized denoised image target. This offers an intuition for why learning $x$ may be easier than learning $u$.
>
> Second, this intuition is consistently supported by our experiments. Both the toy examples and the ImageNet ablations show a clear and stable advantage of modeling the $x$-target over the $u$-target, suggesting that the benefit is not merely heuristic but also practically robust.
>
> In summary, while our current support is not theorem-level, we believe the combination of structural intuition, prior theoretical motivation, and consistent empirical evidence provides a meaningful justification for the learnability advantage of $x(z_t, r, t)$.
>
> **(2) On whether the definition** $x(z_t,r,t)=z_t-t\,u(z_t,r,t)$ **is hand-crafted.**
>
> We appreciate this point. We do not claim that this form is theoretically necessary or unique. Rather, it arises naturally from the geometry of the MeanFlow parameterization. Specifically, $u(z_t,r,t)$ represents the average velocity from time $t$ to $r$. Extrapolating from the current point $z_t$ along this direction back to the zero-noise endpoint gives exactly
>
> $x(z_t,r,t)=z_t - t \cdot u(z_t,r,t).$
>
> So this quantity is not introduced as an arbitrary heuristic, but as the implied clean-endpoint estimate under the flow. This also clarifies why it is natural to interpret $x(z_t,r,t)$ as a generalized denoised-image-like target. That said, we agree that whether this is the *optimal* or *theoretically necessary* parameterization remains open, and we view this as an interesting direction for future work.
>
> In summary, we view $x=z_t-t\cdot u$ as a natural and interpretable reparameterization, rather than an arbitrary hand-crafted choice.
>
> **(3) On the dependence on MeanFlow / iMF.**
>
> We fully agree that pMF is instantiated within the MeanFlow / iMF framework, rather than proposed as a completely independent one-step framework. We will make this scope clearer in the revision.
>
> We view this as a general design perspective, decoupling prediction space from loss space, which is instantiated in this paper on top of MeanFlow/iMF, and that may also inspire related explorations in other domains or formulations.
>
> **(4) On the effect of different pre-conditioners.**
>
> We agree that this is an important point. Our interpretation is not that pre-conditioning is universally harmful, but that in this particular one-step pixel-space regime, the pre-conditioner mixes the target with the noisy input and therefore weakens the intended inductive bias of direct image-like prediction.
>
> Specifically, pre-conditioners change the effective prediction target seen by the network, a pre-conditioned output has the form
>
> $x_\theta = c_{\text{skip}}\cdot z_t + c_{\text{out}}\cdot\mathrm{net}_\theta,$
>
> so unless $c_{\text{skip}}=0$, the direct network output is no longer pure x-space prediction.
>
> As explained before, this matters especially in our very high-dimensional pixel-space regime, where the benefit of pMF comes precisely from making the network output lie as close as possible to an image-like, lower-dimensional target. Empirically, this interpretation is consistent with Table 3(a), where all three pre-conditioned variants are significantly worse than the plain x-prediction formulation.  In future revisions, we will make this explanation more explicit.

---

> > ### Author Rebuttal · Reviewer_SAtk · 2026-04-03
> >
> > Although I still have some reservations about the theoretical support of the paper, I believe this is nevertheless an interesting and thought-provoking work. The perspective it proposes is relatively novel, and the idea of decoupling the prediction space from the loss space offers a fresh angle for one-step pixel-space image generation. In this sense, the paper provides a new and meaningful direction for the image generation community. Overall, despite my remaining questions on the theory side, my overall assessment remains unchanged, and I keep my original score.

---

### Decision · Program_Chairs · 2026-04-30

**Decision:**

Accept (regular)

**Comment:**

In this paper the authors present pixel mean flows (pMFs), which is a simple but effective reparameterization of the original mean flows framework. Normally, in mean flows, a network $u_\theta(z_t, r, t)$ predicts the average velocity over an interval $[r, t]$. In pMFs, the network output is interpreted as an extrapolated data point $x_\theta(z_t, r, t)$. The mean velocity is then defined in terms of this estimate as $u_\theta(z_t, r, t) = (z_t - x_\theta(z_t, r, t)) /t$. Aside from this reparameterization, pixel mean flows are a drop-in replacement for mean flows. The authors do a relatively thorough evaluation which shows good results on ImageNet 256 x 256, whilst also being amenable to generation directly in pixel space (rather than in the latent space of a frozen encoder decoder as in the original MF and follow-up iMF work). Ablations show that, in pixel space, pMF substantially outperforms the MF ablation. The authors hypothesize that this is because estimates $x_\theta(z_t, r, t)$ will be close to the data manifold for most intervals $[r,t]$ which may mean that the network only needs to represent a lower dimensional submanifold, though no concrete analysis is performed to support this hypothesis.

Reviewers are overall supportive of this submission. Given the simplicity of the approach, reviewers note that the technical contribution here has limited novelty, and several reviewers would have liked to have seen the paper offer deeper insights and/or theoretical analysis to explain why this reparameterization is effective. With that said, this appears to be a simple trick that works, which ought to be enough for acceptance. The authors may wish to expand exposition around Eq (8) to make the intuition offered in figure 1 (extrapolation along a mean velocity to time 0) explicit.